# Which Clinical and Biochemical Parameters Are Associated with Lifetime Suicide Attempts in Bipolar Disorder?

**DOI:** 10.3390/diagnostics12092215

**Published:** 2022-09-13

**Authors:** Teresa Surace, Enrico Capuzzi, Alice Caldiroli, Alessandro Ceresa, Cecilia Maria Esposito, Anna Maria Auxilia, Ilaria Tagliabue, Martina Capellazzi, Francesca Legnani, Martina Di Paolo, Luisa Cirella, Francesco Zanelli Quarantini, Maria Salvina Signorelli, Eugenio Aguglia, Massimo Clerici, Massimiliano Buoli

**Affiliations:** 1Department of Mental Health and Addiction, Azienda Socio Sanitaria Territoriale Monza, Via G.B, Pergolesi 33, 20900 Monza, Italy; 2Department of Neurosciences and Mental Health, Fondazione IRCCS Ca’ Granda Ospedale Maggiore Policlinico, Via F, Sforza 35, 20122 Milan, Italy; 3Department of Medicine and Surgery, University of Milano Bicocca, via Cadore 38, 20900 Monza, Italy; 4Healthcare Professionals Department, Foundation IRCCS Ca’ Granda Ospedale Maggiore Policlinico, 20122 Milan, Italy; 5Department of Clinical and Experimental Medicine, Psychiatry Unit, University of Catania, Via Santa Sofia 78, 95123 Catania, Italy; 6Department of Pathophysiology and Transplantation, University of Milan, via F. Sforza 35, 20122 Milan, Italy

**Keywords:** bipolar disorder, suicide attempts, clinical features, biochemical markers

## Abstract

Introduction: Bipolar Disorder (BD) is a disabling condition with suicidal behavior as one of the most common adverse outcomes. The purpose of the present research is to investigate the relationship between lifetime suicide attempts and the clinical factors/biochemical parameters in a large sample of bipolar patients. Methods: A total of 561 patients, consecutively hospitalized for BD in Milan and Monza (Italy), were recruited. Data about the demographic and clinical variables, as well as the values of blood analyses, were collected. The groups identified according to the presence/absence of lifetime suicide attempts were compared using univariate analyses. Then, three preliminary binary logistic regressions and a final logistic regression model were performed to identify the clinical and biochemical parameters associated with lifetime suicide attempts in BD. Results: Lifetime suicide attempts in BD were predicted by a longer duration of untreated illness (DUI) (*p* = 0.005), absence of lifetime psychotic symptoms (*p* = 0.025), presence of poly-substance use disorders (*p* = 0.033), comorbidity with obesity (*p* = 0.022), a last mood episode of manic polarity (*p* = 0.044), and lower bilirubin serum levels (*p* = 0.002); higher total cholesterol serum levels showed a trend toward statistical significance (*p* = 0.058). Conclusions: BD patients with lifetime suicide attempts present unfavorable clinical features. Some specific biochemical characteristics of bipolar patients may represent potential markers of suicidal behavior and need to be better investigated to identify new targets of treatment in the framework of personalized medicine. These preliminary findings have to be confirmed by further studies in different clinical settings.

## 1. Introduction

Suicide is a serious global public health issue [1]. It is defined as the act of intentionally taking one’s own life and it is among the ten leading causes of death worldwide, with more deaths attributable to it than breast cancer, war, or homicide [2]. Indeed, nearly 800,000 people die by suicide every year [2]. Suicide attempts have become more prevalent in recent years, with notable increases in different geographical areas for all age groups [3]. Suicidal behavior is a complex phenomenon involving environmental factors and biological vulnerability [4]. Of note, the available literature points out that different biological factors predispose a person to attempted suicide including epigenetic modifications [5] and abnormalities in lipid metabolism [6,7,8,9], inflammation [10], monoamine neurotransmission [11], and the hypothalamic–pituitary–adrenal (HPA) axis [12]. On the other hand, suicide and the related biological changes may be triggered by proximal (e.g., the onset of a psychiatric disorder) or distal events such as childhood trauma [13,14].

Mood and psychotic disorders represent one of the most important risk factors for suicide [15]. The scientific literature shows that mental illness could represent a heavy risk factor, with up to 90% of people who commit suicide suffering from a psychiatric condition [16]. Furthermore, post-mortem studies indicate that up to two-thirds of all deaths from suicide are committed by people who suffer from mood disorders, the first among them being Bipolar Disorder (BD) [17]. In addition, the risk of suicide attempts in bipolar patients was calculated to be 20–30 times higher than that observed in the general population [18].

The presence of lifetime suicide attempts reflects a more severe course of BD [19]. The first meta-analysis by the International Society for BD reported that different clinical features predict suicide attempts in BD including the depressive polarity of the current or most recent episode and comorbidity with substance-use disorders or borderline personality disorder [20]. Of note, a recent multicentric large-sample Italian study identified an earlier age at first contact with psychiatric services and the presence of psychotic symptoms or hospitalizations in the last year among the factors associated with suicidal behavior in bipolar patients [21]. On the other hand, the identification of the clinical variables predicting self-harm in BD is extremely important to establish proper treatment with anti-suicidal compounds such as lithium [22,23]. Therefore, many studies have been recently undertaken to better understand the pathophysiology of suicide attempts in BD as well as to identify potential peripheral markers of suicidal behavior, with the scope of developing preventive strategies and identifying effective drugs. Even though no specific set of neurobiological markers associated with the risk of suicide attempts has been identified for people with BD, the potential biological correlates have been recognized. For instance, increasing evidence shows the important role of cell-mediated immune activation and chronic inflammation both in the pathophysiology of BD and in the risk of suicidal behavior. In this regard, previous studies reported abnormal levels of pro-inflammatory factors, including interleukin-6, tumor necrosis factor, and C-reactive protein (CRP) [24], in patients with BD and suicidal behavior. Moreover, the neutrophil–lymphocyte ratio (NLR) was suggested as a convenient and reproducible biomarker of suicide attempt risk in BD [25]. Nevertheless, some evidence demonstrates the simultaneous role of inflammation and oxidative stress in conferring suicidal risk in bipolar patients as shown by increased lipid peroxidation, insulin resistance, oxidative damage of membranes, and DNA as well as mitochondrial dysfunction [4]. Indeed, some antioxidant agents, including uric acid, albumin, and high-density lipoproteins, were found to be lower among patients affected by mood disorders than controls, whereas some oxidative damage products, including malondialdehyde levels, were found to be higher in patients than controls [26]. Of note, bilirubin being an endogenous antioxidant may be impaired by excessive oxidative stress. It is not surprising that low total serum bilirubin levels were proposed as a risk factor for different neurological and psychiatric disorders including BD [27]. In the same way, obesity concomitant with over-inflammation has led some authors to argue that metabolic syndrome and the related metabolic biomarkers may be related to suicide attempts in BD but so far, available data are inconsistent [28]. Similarly, even though it has been suggested that chronically elevated cortisol levels, resulting from a pro-inflammatory state can potentially be associated with suicidal behavior, the findings appear inconsistent [29]. Nevertheless, since lipids are involved in important brain functions, including cell membrane integration, regulation of synapses, energy metabolism, and neuroendocrine regulation, changes in the plasma levels of lipids may contribute to the risk of self-harm in BD [30]. Despite the available literature, a global model integrating all factors associated with suicidal attempts in BD is lacking. Of note, few studies have concomitantly evaluated the different clinical variables and laboratory markers in bipolar patients in relation to suicidal risk. Thus, in order to gain more data for clinical practice and to detect any specific features associated with suicidal behavior in BD, the purpose of this study was to explore the possible roles of several clinical variables and routine biochemical markers in the lifetime occurrence of suicide attempts in a sample of real-world inpatients affected by BD.

Therefore, our main research questions were:What is the prevalence of lifetime suicide attempts in hospitalized patients with BD?Are there some clinical and biological variables for discriminating between BD individuals with versus without lifetime suicide attempts?Which of these variables are related to each other?Is it possible to hypothesize some interactions between the different biological pathways associated with suicide attempts?

Given the exploratory nature of our clinical and biomarker panel, we expected that BD patients with a history of lifetime suicide attempts may differ from their counterparts for some of the included variables. These results could be helpful for implementing preventative medicine in order to provide individualized strategies for the management and treatment of suicidal risk in BD patients.

## 2. Methods

This is a retrospective cross-sectional study, with information mainly obtained from chart reviews or intranet hospital applications (for the values of blood analyses). An overall sample of 561 patients with a diagnosis of BD according to DSM criteria (American Psychiatry Association, 2013) among patients consecutively admitted to inpatient service of Fondazione IRCCS Policlinico, Milan, Italy (N = 522), and ASST Monza, Italy (N = 39), was enrolled. The enrollment covered the period 2008–2021 for the first center and 2019–2021 for the second center, explaining the difference in numbers. In the case of multiple hospitalizations, the last one was considered for the present analysis. The diagnosis of BD, as well as the main psychiatric comorbidities including personality disorders, was made by an expert senior psychiatrist from the inpatient clinic staff through a clinical interview according to the DSM-5 criteria. Previous research by our group showed a high rate of agreement between the clinical diagnosis and the use of diagnostic tools for the diagnosis of mood disorders [31]. Clinical and biological data were obtained through a screening of the clinical charts, intranet hospital applications (blood analyses conducted during the first day of hospitalization), and interviews with patients and their relatives in the rare case that the information was not available in the charts. Of note, almost all the patients hospitalized for BD were then followed up in the local community mental health services. The inclusion criteria consisted of (1) having an age ≥18 and (2) being hospitalized for a mood episode in the context of a diagnosis of BD. Exclusion criteria were the following: (1) an age ≤18, (2) peripartum as perinatal mood disorders (with onset during pregnancy or one month after delivery) presents a different clinical presentation associated with specific biological abnormalities [32]; (3) pharmacological treatments that could significantly change the mood (steroids, levetiracetam, interferon, efavirenz); (4) medical comorbidities that could significantly affect mood or biochemical parameters (e.g., active rheumatoid arthritis or multiple sclerosis). In the case of psychiatric comorbidities, BD represented the disorder that affected the patients for a longer time or/and was responsible for more disability. The protocol of this study was approved by the Fondazione IRCCS Ca’ Granda Ospedale Maggiore Policlinico Ethical Committees (approval number 1789) and it conformed to the provisions of the Declaration of Helsinki.

The following variables were collected on the first day of hospitalization:

Clinical variables (lifetime): age at onset, duration of untreated illness (DUI) (years), duration of illness (years), number of previous hospitalizations, number of lifetime major mood episodes, number of lifetime manic episodes, number of lifetime depressive episodes, number of lifetime hypomanic episodes, lifetime presence of mood episodes with mixed features, lifetime psychotic symptoms, bipolar subtype, gender, lifetime presence of rapid cycling, lifetime presence of seasonality, family history of psychiatric disorders, presence of multiple family history of psychiatric disorders, presence and type of lifetime substance-use disorders, presence of lifetime poly-substance-use disorders, comorbid personality disorders, psychiatric comorbidity, medical comorbidity, presence of multiple medical comorbidities, history of obstetrical complications.

Clinical variables (current or last year): age at hospitalization, duration of hospitalization (days), smoking status, number of cigarettes/day, number of major mood episodes in the last year, number of manic episodes in the last year, number of depressive episodes in the last year, number of hypomanic episodes in the last year, number of major mood episodes triggered by substance-use disorder in the last year, type of current episode (mania/depression), presence of mixed features, type of last major mood episode, administration of poly-therapy during the last mood episode, comorbidity with hypothyroidism, diabetes, hypercholesterolemia, obesity, achievement of treatment response/remission during the current hospitalization, current treatment with statins and type of statin, current treatment with levothyroxine, and scores of Global Assessment of Functioning (GAF), Young Mania Rating Scale (YMRS), Hamilton Depression Rating Scale (HAM-D), Montgomery–Asberg Depression Rating Scale (MADRS), Brief Psychiatric Rating Scale (BPRS), Hamilton Anxiety Rating Scale (HAM-A).

Biochemical parameters (measured on blood or serum): number of white blood cells (WBC) (10^9^/L), number of red blood cells (10^12^/L), hemoglobin (HB) (g/dL), mean corpuscular volume (MCV) (fL), number of platelets (10^9^/L), mean platelet volume (MPV) (fL), number of neutrophils (10^9^/L), number of lymphocytes (10^9^/L), NLR, glycaemia (mg/dL), blood urea (mg/dL), creatinine (mg/dL), uric acid (mg/dL), aspartate aminotransferase (AST) (U/L), alanine aminotransferase (ALT) (U/L), gamma-glutamyltransferase (GGT) (U/L), total bilirubin (mg/dL), total plasmatic proteins (g/dL), albumin (g/dL), lactate dehydrogenase (LDH) (U/L), creatine phosphokinase (CPK) (U/L), pseudocholinesterase (PCHE) (U/L), total cholesterol (mg/dL), serum iron (mcg/dL), thyroid-stimulating hormone (TSH) (mcU/mL), CRP (mg/dL).

The biochemical parameters represented the universal screening for patients admitted to the two hospitals. In some cases, the number of missing values was high as the screening panels at admission were slightly different in the two inpatient clinics.

Concerning the rating scales, GAF measures patients’ global functioning, with total scores ranging from 1 to 100 and higher scores indicating less impairment [33]. YMRS is a tool used to assess the severity of mania, with a score ≥10 indicating clinically significant symptoms [32]. Both HAM-D and MADRS assess the severity of depressive symptoms, with the former focusing more on the anxiety–somatization aspects (HAM-D ≥8 indicating clinically significant depressive symptoms) [34] and the latter focusing more on identifying the core symptoms of major depression (e.g., guilt feelings and anhedonia). MADRS scores ≥10 reveal clinically significant depressive symptoms (Hawley et al., 2002). BPRS measures the severity of general psychopathology, with a score ≥31 indicating the presence of clinically significant psychiatric symptoms [35], and HAM-A assesses the severity of anxiety symptoms, with a score ≥8 requiring clinical attention [36]. The rating scales are routinely administered at the inpatient clinic of Fondazione IRCCS Policlinico, Milan, which is different from Monza, and for this reason, this information was missing for all patients from this site.

DUI was considered as the time between the first major mood episode of BD and the prescription of proper pharmacological treatment (mood stabilizer or atypical antipsychotic with stabilizing effects) [37]. The presence of mixed features was established according to the DSM-5 criteria as in the previous research by our group [38]. A suicide attempt was defined as self-harm combined with the intent to die. Self-harm without intent was not taken into account [21]. Obesity was defined as a body mass index (BMI) ≥30 [39]. Treatment response was defined as a reduction in the YMRS or HAM-D scores ≥50%, and remission as a YMRS score <10 and a HAM-D score <8, similar to the previous researches performed by our group [38,40] and according to the previous literature [41,42].

Statistical analyses were performed using The Statistical Package for Social Sciences (SPSS) for Windows (version 27.0). Descriptive analyses of the total sample were initially performed and then the two groups identified according to the presence of lifetime suicide attempts were compared using Student’s *t*-tests for quantitative variables and χ^2^ tests for qualitative ones. Subsequently, the statistically significant variables from these preliminary analyses were inserted into binary logistic regression models (enter method) using the presence of lifetime suicide attempts as the dependent variable. Three different binary logistic regression analyses were computed, one for the lifetime clinical variables, one for the recent (last year) or current clinical variables, and one for the biochemical parameters.

Every statistically significant variable from the three above-mentioned binary logistic regression models was inserted into a new global starting multivariable logistic regression model (enter method) in order to identify the variables independently associated with the presence of lifetime suicide attempts. The goodness of the models was assessed by the Omnibus and Hosmer–Lemeshow tests. The level of statistical significance was set at *p* ≤ 0.05. A similar statistical approach has been used by our group in other analyses of different samples due to the supposed large number of variables statistically related to the dependent variable (in this case the presence of lifetime suicide attempts) [43]. In the final model, we did not insert all the variables to limit the risk of multicollinearity and to better identify the variables that are the potential predictors of lifetime suicide attempts.

Finally, univariate analyses were performed to compare the BD I and BD II subgroups in terms of the presence/absence of lifetime suicide attempts and the variables that were inserted into the final regression model.

## 3. Results

The total sample included 561 patients, of whom 220 were male (39.2%) and 341 female (60.8%). The mean age was 47.34 ± 14.77 years. The mean number of lifetime suicide attempts in the group of suicide attempters was 1.58 ± 1.09. Descriptive analyses and the results of univariate analyses are reported in Table 1 (lifetime clinical variables), Table 2 (current or last year clinical variables), and Table 3 (biochemical parameters). With regard to the main pharmacological treatment before hospitalization, the information was available and reliable for 259 patients. Four patients were not pharmacologically treated, and the remainder were treated as follows: 62 with valproic acid, 64 with lithium, 3 with risperidone, 16 with haloperidol, 6 with paliperidone, 24 with olanzapine, 24 with quetiapine, 11 with aripiprazole, 3 with lamotrigine, 1 with asenapine, 9 with selective serotonin reuptake inhibitors (SSRIs), 11 with serotonin-norepinephrine reuptake inhibitors (SNRIs), 4 with zuclopenthixol, 3 with gabapentin, 3 with carbamazepine, 4 with vortioxetine, 2 with mirtazapine, 2 with trazodone, 2 with tricyclic antidepressants, and 1 with lurasidone. No statistically significant differences were found according to the type of prescribed pharmacotherapy among the two groups identified by the presence of lifetime suicide attempts (χ^2^ = 15.78, *p* = 0.78).

Univariate analyses showed that patients with lifetime suicide attempts (compared to those without suicidal behavior) had an earlier age at onset (t = 3.587, *p* < 0.001); current lower GAF scores (t = 3.118, *p* = 0.002); a longer DUI (t = 2.064, *p* = 0.041) and DI (t = 4.414, *p* < 0.001); a higher number of total (t = 2.982, *p* = 0.003), hypomanic (t = 3.220, *p* = 0.001) and depressive (t = 4.466, *p* < 0.001) lifetime episodes; a higher number of total mood episodes (t = 2.182, *p* = 0.030) and depressive episodes in the last year (t = 3.322, *p* = 0.001); lower current YMRS total scores (t = 4.005, *p* < 0.001); and lower bilirubin serum levels (t = 4.181, *p* < 0.001) but higher total cholesterol serum levels (t = 2.021, *p* = 0.044). In addition, patients with lifetime suicide attempts (with respect to those with no history of self-harm) were found to be more frequently hospitalized for a current depressive episode (χ^2^ = 26.788, *p* < 0.001, OR = 3.135), to present more frequently current mixed features (χ^2^ = 4.384, *p* = 0.045, OR = 1.567) and a history of poly-substance-use disorders (χ^2^ = 5.270, *p* = 0.024, OR = 1.868), to experience fewer lifetime psychotic symptoms (χ^2^ = 6.638, *p* = 0.012, OR = 0.579), to be predominantly women (with a trend toward statistical significance, χ^2^ = 3.873, *p* = 0.056, OR = 1.541), to have more frequent multiple family histories for psychiatric disorders (with a trend toward statistical significance, χ^2^ = 3.706, *p* = 0.063, OR = 1.632) or a history of heroin-use disorders (5.5% versus 0.3%) among the different substances of abuse (χ^2^ = 22.021, *p* = 0.005), to have experienced more often a last mood episode of manic polarity (χ^2^ = 10.398, *p* = 0.018), to suffer more frequently from a personality disorder (χ^2^ = 17.371, *p* < 0.001, OR = 3.148) and more specifically a borderline personality disorder (15.5% versus 4.0%), to have suffered more often from an eating disorder (14.6% versus 2.2%) among the different psychiatric comorbidities (χ^2^ = 12.376, *p* = 0.014), to have more frequent multiple medical comorbidities (χ^2^ = 6.848, *p* = 0.011, OR = 1.976), to suffer more often from obesity (with a trend toward statistical significance, χ^2^ = 3.779, *p* = 0.059, OR = 2.171), and to present more common lifetime rapid cycling (χ^2^ = 8.069, *p* = 0.006, OR = 2.237). No statistically significant differences were found in the other analyzed variables (*p* > 0.05).

With regard to the preliminary binary logistic regression model with the lifetime clinical variables, the goodness-of-fit test results (Hosmer and Lemeshow Test: χ^2^ = 12.322, *p* = 0.137) showed that the model was reliable, allowing for a correct classification of 83.0% of the cases. In addition, the model was significant overall (Omnibus test: χ^2^ = 50.726, df = 29, *p* = 0.008). Patients with a history of suicide attempts were found to have a longer DUI (*p* = 0.005, OR = 1.204, confidence interval (CI) = 1.057–1.371), to present less frequent lifetime psychotic symptoms (*p* = 0.025, exponential B (expB) = 4.376, CI = 1.208–15.859), and to have a more frequent history of poly-substance-use disorders (*p* = 0.033, expB = 0.058, CI = 0.004–0.798). Concerning the preliminary binary logistic regression model with the current or recent clinical variables (last year), the goodness-of-fit test results (Hosmer and Lemeshow test: χ^2^ = 7.507, *p* = 0.483) showed that the model was reliable, allowing for a correct classification of 75.0% of the cases. In addition, the model was significant overall (Omnibus test: χ^2^ = 19.050, df = 10, *p* = 0.040). Patients with a history of suicide attempts were found to be more frequently affected by obesity (*p* = 0.022, expB = 0.270, CI = 0.088–0.830) and have experienced a last mood episode of manic polarity (*p* = 0.044). Finally, regarding the preliminary logistic regression model with the biochemical parameters, the goodness-of-fit test results (Hosmer and Lemeshow test: χ^2^ = 13.423, *p* = 0.098) showed that the model was reliable, allowing for a correct classification of 77.2% of the cases. In addition, the model was significant overall (Omnibus test: χ^2^ = 17.148, df = 2, *p* < 0.001). Patients with a history of suicidal behavior were confirmed to have lower bilirubin serum levels (*p* = 0.002, OR = 0.160, CI = 0.050–0.507) as well as higher total cholesterol serum levels with a trend toward statistical significance (*p* = 0.058, OR = 1.006, CI = 1.000–1.012).

The final model was found to be reliable (Hosmer and Lemeshow test: χ^2^ = 9.654, *p* = 0.290), allowing for a correct classification of 80.0% of the cases. In addition, the model was significant overall (Omnibus test: χ^2^ = 17.342, df = 8, *p* < 0.027). Patients with lifetime suicide attempts were found to have a more frequent history of poly-substance-use disorders (*p* = 0.003) and a longer DUI with a trend toward statistical significance (*p* = 0.087) (Table 4). We performed a correlation analysis to identify the eventual relationships among the quantitative variables (total cholesterol, bilirubin, and DUI) and the analysis revealed a weak inverse correlation between bilirubin and cholesterol (r = −0.109, *p* < 0.05).

Finally, the BD I and BD II patients were not statistically different in terms of the presence/absence of lifetime suicide attempts (χ^2^ = 2.113, *p* = 0.146, OR = 2.511). Moreover, DUI (t = 0.825, *p* = 0.410), total bilirubin serum levels (t = 0.814, *p* = 0.416), total cholesterol serum levels (t = −0.248, *p* = 0.804), history of poly-substance-use disorders (χ^2^ = 0.259, *p* = 0.611, OR = 0.586), and obesity (χ^2^ = 0.165, *p* = 0.684, OR = 0.995) were found to be similar in BD I and BD II. The two diagnostic subgroups were significantly different for the presence of lifetime psychotic symptoms (χ^2^ = 12.964, *p* < 0.01, OR = 0.059) and the type of the last mood episode (χ^2^ = 15.907, *p* = 0.001); in particular, BD I patients reported more frequent lifetime psychotic symptoms than BD II patients (*p* < 0.05), and the BD II subgroup experienced more frequent hypomania as the last mood episode than their counterparts (*p* < 0.05).

## 4. Discussion

### 4.1. Main Findings

This is one of the few studies evaluating a large set of clinical and biochemical variables in relation to suicide attempts in a sample of bipolar inpatients. We found that 21% of bipolar individuals consecutively admitted to two Italian hospitals attempted suicide at least one time in the course of their lives, a lower value in comparison with a relatively recent meta-analysis reporting a rate of 33.9% [44]. Our figures could be lower as a result of the specific inclusion/exclusion criteria of our study, different patterns of comorbidity, and the fact that the involved hospitals are academic centers specializing in the treatment of mood disorders. Nevertheless, our results are consistent with prior reports and highlight the strong association between BD and suicidal behavior [45].

The univariate analysis showed that a lifetime presence of suicide attempts was related to a substantial number of demographic and clinical variables, whereas the binary logistic regression models showed that a history of suicide attempts was associated with longer DUI, less frequent lifetime psychotic symptoms, a history of poly-substance-use disorders, obesity, a last mood episode of manic polarity, lower bilirubin serum levels, and higher total cholesterol serum levels, compared to their counterparts. A history of poly-substance-use disorders and longer DUI, although showing a trend toward statistical significance, were also confirmed as variables associated with suicidal behavior in the final regression model. Although some of these findings replicated the previous literature, some other findings were unprecedented.

With regard to the clinical variables, our results seem to confirm those of prior studies, specifically showing that suicide attempts in BD may be predicted by longer DUI. Altamura and co-authors [46] reported that among patients with more than two years of untreated illnesses, the frequency of suicide attempters and suicide attempts was significantly higher compared to those with 2 years or less of untreated BD. However, some considerations should be taken into account. First, BD is still underdiagnosed, undertreated, and often mistreated so much that there is often a 5–10-year delay for proper diagnosis and management, especially in the long term [3]. Second, a longer DUI is a predictor of unfavorable outcomes in BD, not only for the increased risk of suicide attempts but also because it is related to a longer duration of illness, higher frequency of comorbidity with panic disorders and substance abuse, as well as higher frequencies of courses of rapid cycling hospitalization [46]. Third, DUI may be a predictor of worse outcomes, especially among bipolar patients with psychotic features since they may frequently receive a different diagnosis at first contact with psychiatric services [47]. However, we found that bipolar subjects with lifetime suicide attempts were found to have less frequent lifetime psychotic symptoms. At present, there is a lack of agreement on whether the presence of psychotic symptoms is a risk factor for suicidal behavior. Indeed, data regarding a possible association between psychotic symptoms and suicidal behavior are inconsistent, although an association between psychotic depression and suicidal behavior in BD has been hypothesized. When compared with psychotic mania or mixed episodes, Dell’Osso and co-authors [48] found that psychotic bipolar depression was associated with lifetime suicidal ideation and behavior. In contrast, two studies reported no differences in psychotic symptoms between bipolar subjects who attempted suicide and those who did not [49,50]. Furthermore, Azorin and co-authors [51] found that inpatients with mania and a history of suicide attempts reported less frequent prominent psychotic symptoms than subjects without suicidal behavior. In agreement, Oquendo and co-authors [52] reported that those who attempted suicide had a trend toward fewer psychotic symptoms compared to those who did not attempt suicide, assuming that psychotic symptoms might affect the ability to plan and execute self-harm. In addition, our results highlighted that recent manic episodes may have an important role in conferring vulnerability to suicide attempts in bipolar patients. Even though most studies reported a higher likelihood of suicide attempts during current depressive and mixed states [13,20], some authors argued a possible association of self-harm with manic episodes with mixed features [53]. In addition, a relationship between lifetime suicide attempts and antidepressant-induced mania has been reported [54], which, together with a high number of depressive episodes, early age at onset, and comorbid substance-use disorder, could identify a subgroup of bipolar patients with a high suicide risk [55]. Indeed, patients with a depressive polarity of first episodes may be misdiagnosed and therefore treated with only antidepressant mono-therapy, resulting in poor prognosis in terms of inadequate stabilization, higher impulsivity, and occurrence of suicide attempts [21]. Finally, we can hypothesize that some patients may experience a manic episode as part of the rapid-cycling course of BD, which is a major risk factor for suicide attempts in BD [19,36]. 

In agreement with the previous literature, patients with lifetime suicidal behavior may present a more frequent history of poly-substance-use disorders. According to a meta-analysis involving 7952 individuals, bipolar individuals with lifetime comorbid substance-use disorders had a higher risk of suicidal behaviors compared with those without substance-use disorders (OR = 1.77), with no significant heterogeneity across studies [56]. Particularly, this significant relationship was observed with both alcohol-use disorder and other substance-use disorders, especially in the case of prolonged substance misuse. It is also noteworthy that substance-use disorders may be a BD-type-I-specific correlate of suicidal behavior [57], unlike BD type II [58]. However, multiple factors should be considered in interpreting the association between suicide attempts and substance-use disorders in bipolar patients. Particularly, some authors argued a potential role of aggressive-impulsive traits in the association between substance-use disorders and suicide attempts in BD [59]. Nevertheless, genetic factors, a family history of BD, childhood trauma, tobacco smoking, early onset of BD, and comorbid anxiety disorders may also potentiate the possibility of self-harm among bipolar patients with a history of poly-substance-use disorders [54,59,60,61]. On the other hand, substance misuse favors mood instability and impulsivity with an increased risk of suicidal behavior in patients affected by BD [62]. Finally, given that substances of abuse amplify the biological abnormalities and cognitive impairments associated with BD, bipolar patients with concomitant substance-use disorders potentially suffer from more severe variants of the disorder with a consequently high suicidal risk [63,64].

Our results showed that obesity contributes to an increase in the risk of suicide among bipolar patients. Previous data from clinical samples showed that the prevalence of obesity in BD patients was higher than in the general population, ranging from 20 to 35% of bipolar individuals [65]. In addition, more than half of patients with BD become obese within 20 years after their first hospitalization [66]. However, data about a potential relationship between lifetime suicide attempts and obesity appear controversial [67]. Some studies reported significantly higher percentages of suicide attempts among obese versus non-obese bipolar patients [68,69]. However, other studies failed to confirm these findings [70]. On the other hand, different studies showed a robust association between obesity and violent suicide attempts [71,72]. Some authors suggested that some clinical features, such as rapid cycling and depression with atypical features, can explain the association between obesity and suicide attempts in BD [69,73]. Moreover, obesity seems to worsen the cognition of bipolar patients, particularly their verbal memory and frontal abilities, with a more pronounced effect on violent suicide attempters [72]. Indeed, a higher BMI may be associated with brain white-matter abnormalities, especially in the networks that regulate mood and impulsivity [74,75].

With regard to cholesterol, available data in the literature are controversial about the association of this biochemical parameter with the risk of suicide attempts in BD. A meta-analysis by Bartoli and colleagues, which included 1042 individuals affected by BD, found no differences in total cholesterol and triglyceride profiles between suicide attempters and non-attempters. To date, a significant number of observational studies exploring the association between lipid serum levels and suicide attempts in subjects with BD have produced mixed results, also taking into account the high methodological heterogeneity across the studies [8,76]. However, in agreement with our findings, the important link between obesity, metabolic syndrome, and hypercholesterolemia, which in turn is associated with a pro-inflammatory state directly affecting the brain, should be considered [77]. Even though some studies failed to find an association between suicide attempts and metabolic syndrome in bipolar patients [78], there are authors who suggested a specific role of abdominal fat deposits [72]. Of note, abdominal obesity is associated with an increased HPA axis responsivity and increased expression of glucocorticoid receptors in adipose tissue, possibly further sustaining the vicious cycle of obesity, impaired stress response, and increased levels of total and low-density lipoproteins [79]. Abdominal adipose tissue expansion, in turn, can promote low-grade inflammation via the local production of pro-inflammatory cytokines [4]. Particularly, high levels of pro-inflammatory cytokines and HPA axis hyper-reactivity were related to a history of violent suicide attempts among bipolar patients [55]. Moreover, a number of data support that over-inflammation, increased oxidative stress, and altered lipid profiles are interrelated phenomena explaining aggressive behavior or suicide attempts in BD [80,81,82,83,84].

Finally, a cross-sectional study found that the severity of DNA damage might be the only significant factor that differentiated suicide attempters from non-attempters among BD patients with lifetime suicide ideation [85]. Bilirubin, as an endogenous antioxidant, may therefore be impaired by excessive oxidative stress [43]. Furthermore, it is not surprising that lower total serum bilirubin levels were proposed as a vulnerable risk for different neurological and psychiatric diseases including BD [86]. In the same way, as shown by our findings, an inverse correlation between bilirubin and total cholesterol supports the role of dyslipidemia and related inflammation in compromising oxidative defenses in bipolar patients, thus resulting in increased biological vulnerability to severe clinical forms of BD [85].

From a practical point of view clinicians:(1)should verify the presence of substance-use disorders or metabolic abnormalities in subjects with BD for the association of these characteristics with suicidal behavior.(2)could choose to treat bipolar patients with past suicidal attempts with compounds that have a low impact on metabolism and poor interaction with the cytochrome system.(3)could apply psychoeducational approaches to favor healthy lifestyles such as a diet poor in saturated fatty acids or physical activity to ameliorate the outcomes of patients affected by BD.

### 4.2. Limitations

Methodological limitations have to be taken into account in the interpretation of the results of the present article. First, as this study was a cross-sectional one, we cannot definitively define whether the selected variables preceded or followed the occurrence of a suicide attempt. Moreover, the data were collected retrospectively so they can be imprecise in some cases. In addition, with regard to some of the variables, a lot of data are missing because some of the biochemical parameters were not routinely collected at the admission of patients in one or both of the hospitals involved in this study. Missing data are an important limitation when interpreting results. Nevertheless, missing values may result in a loss of statistical power [86]. Second, we analyzed lifetime suicide attempts. Our sample, therefore, does not include completed suicides and data refer only to individuals who survived the suicide attempt. Third, we should consider that most subjects were treated with pharmacotherapy, potentially affecting several biochemical parameters [4]. Nevertheless, some drugs may have promoted the occurrence of mood episodes (for instance, the prescription of first-generation antipsychotics during mania favoring the switch to depression or the development of mania or hypomania during antidepressant treatment). However, no statistically significant differences were found between the groups according to the type of prescribed treatment before hospitalization. Fourth, we cannot fully rule out that part of our sample may be affected by subclinical conditions (e.g., cardiovascular, autoimmune diseases, infections), which in turn may translate into the alteration of some biochemical parameters [87]. Fifth, we did not perform statistical analyses considering single versus multiple lifetime suicide attempts as well as violent versus non-violent suicide attempts. Of note, bipolar patients with multiple suicide attempts may present peculiar socio-demographic features and clinical characteristics compared with patients with a single suicide attempt [65]. Similarly, some biochemical parameters such as cholesterol and NLR may differentiate patients with violent versus non-violent suicide attempts [8]. Finally, we did not assess some factors, such as childhood trauma, the severity of impulsiveness, and cognitive impairment, that could contribute to an increase in the risk of suicidal behavior in bipolar patients [72].

Despite these limitations, the large sample size and the assessment of a large set of clinical and biochemical variables may represent the strengths of the present study. In addition, individuals enrolled in this study can be considered representative of “real-world” inpatients with BD. However, it is clear that further studies are needed for the generalization and confirmation of these results.

## 5. Conclusions

In agreement with the available literature, a significant number of bipolar patients included in our sample have a history of lifetime suicide attempts. Suicide attempts appear to be associated with unfavorable characteristics including a longer DUI and history of poly-substance abuse. Furthermore, patients with suicidal behavior may present more frequent absences of lifetime psychotic symptoms and last mood episodes of manic polarity. These factors could be taken into account in the assessment of suicidal behavior on admission to hospital. Nevertheless, the presence of obesity, as well as having lower bilirubin serum levels and higher total cholesterol serum levels, may be associated with an increased risk of self-harm. Future prospective studies are needed to confirm our findings in order to better clarify the role of some biological pathways in the onset of suicidal behavior. The findings of this research will favor the identification of a new target for treatments for the better management of bipolar patients with different clinical characteristics such as suicidal behavior [85,88].

## Figures and Tables

**Table 1 diagnostics-12-02215-t001:** Lifetime clinical variables of the total sample and the two groups obtained according to the presence of lifetime suicide attempts.

Variables			Total Sample N = 561	Absence of Lifetime Suicide Attempts N = 443 (79.0%)	Presence of Lifetime Suicide Attempts N = 118 (21.0%)	χ^2^/t	*p*
**Gender**		**Male**	220 (39.20%)	183 (41.3%)	37 (31.4%)	3.873	0.056
		**Female**	341 (60.80%)	260 (58.7%)	81 (68.6%)		
**Age at onset of BD** Missing: *n* = 34			29.29 (±11.43)	30.18 (±11.52)	26.10 (±10.55)	3.587	**<0.001**
**Number of previous hospitalizations** Missing: *n* = 106			3.23 (±4.30)	3.01 (±3.87)	4.02 (±5.52)	1.702	0.091
**Number of lifetime major mood episodes** Missing: *n* = 88			6.09 (±5.41)	5.57 (±4.46)	7.92 (±7.67)	2.982	**0.003**
**Number of lifetime manic episodes** Missing: *n =* 87			2.51 (±3.41)	2.36 (±2.65)	3.04 (±5.27)	1.276	0.205
**Number of lifetime depressive episodes** Missing: *n =* 89			2.23 (±2.04)	2.02 (±1.95)	3.01 (±2.17)	4.466	**<0.001**
**Number of lifetime hypomanic episodes** Missing: *n =* 88			1.34 (±1.90)	1.19 (±1.77)	1.86 (±2.24)	3.220	**0.001**
**Duration of illness (years)** Missing: *n =* 34			17.94 (±13.05)	16.64 (±12.73)	22.61 (±13.14)	4.414	**<0.001**
**Duration of untreated illness (years)** Missing: *n =* 135			3.02 (±5.38)	2.70 (±5.10)	4.16 (±6.20)	2.064	**0.041**
**Lifetime presence of mood episodes with mixed features** Missing: *n =* 121	** *No* **		193 (43.9%)	157 (45.4%)	36 (38.3%)	1.504	0.242
	** *Yes* **		247 (56.1%)	189 (54.6%)	58 (61.7%)		
**Lifetime presence of rapid cycling** Missing: *n =* 93	** *No* **		404 (86.3%)	323 (88.7%)	81 (77.9%)	8.069	**0.006**
	** *Yes* **		64 (13.7%)	41 (11.3%)	23 (22.1%)		
**Lifetime presence of seasonality** Missing: *n =* 155	** *No* **		378 (93.1%)	297 (92.5%)	81 (95.3%)	0.804	0.475
	** *Yes* **		28 (6.9%)	24 (7.5%)	4 (4.7%)		
**History of obstetrical complications** Missing: *n =* 79	** *No* **		473 (98.1%)	371 (98.4%)	102 (97.1%)	0.718	0.416
	** *Yes* **		9 (1.9%)	6 (1.6%)	3 (2.9%)		
**Family history of psychiatric disorders** Missing: *n =* 149	** *None* **		208 (50.5%)	168 (51.4%)	40 (47.0%)	6.931	0.642
	** *MDD* **		62 (15.0%)	48 (14.7%)	14 (16.6%)		
	** *BD 1* **		54 (13.1%)	43 (13.1%)	11 (12.9%)		
	** *BD 2* **		19 (4.6%)	13 (4.0%)	6 (7.1%)		
	** *Anxiety Disorders* **		5 (1.2%)	3 (0.9%)	2 (2.3%)		
	** *Schizophrenia* **		25 (6.1%)	23 (7.0%)	2 (2.3%)		
	** *Eating Disorders* **		3 (0.7%)	2 (0.6%)	1 (1.2%)		
	** *Other psychiatric disorders* **		36 (8.8%)	27 (8.3%)	9 (10.6%)		
**Presence of multiple family histories of psychiatric disorders** Missing: *n =* 160	** *No* **		278 (69.3%)	227 (71.6%)	51 (60.7%)	3.706	0.063
	** *Yes* **		123 (30.7%)	90 (28.4%)	33 (39.3%)		
**Presence of lifetime substance-use disorders** Missing: *n =* 50	** *No* **		321 (62.8%)	257 (64.1%)	64 (58.2%)	1.290	0.267
	** *Yes* **		190 (37.2%)	144 (35.9%)	46 (41.8%)		
**Type of lifetime substance-use disorders** Missing: *n =* 56	** *None* **		320 (63.4%)	256 (64.6%)	64 (58.7%)	22.021	**0.005**
	** *Alcohol* **		62 (12.3%)	47 (11.9%)	15 (13.8%)		
	** *Cocaine* **		32 (6.3%)	22 (5.5%)	10 (9.2%)		
	** *Cannabis* **		76 (15.0%)	63 (15.9%)	13 (11.9%)		
	** *Heroin* **		7 (1.4%)	1 (0.3%)	6 (5.5%)		
	** *LSD* **		1 (0.2%)	1 (0.3%)	0 (0.0%)		
	** *Amphetamine* **		3 (0.6%)	2 (0.5%)	1 (0.9%)		
	** *MDMA* **		2 (0.4%)	2 (0.5%)	0 (0.0%)		
	** *Others* **		2 (0.4%)	2 (0.5%)	0 (0.0%)		
**Presence of lifetime poly-substance-use disorders** Missing: *n =* 61	** *No* **		424 (84.8%)	340 (86.7%)	84 (77.8%)	5.270	**0.024**
	** *Yes* **		76 (15.2%)	52 (13.3%)	24 (22.2%)		
**BD type** Missing: *n =* 41	** *1* **		510 (98.1%)	403 (98.5%)	107 (96.4%)	2.113	0.146
	** *2* **		10 (1.9%)	6 (1.5%)	4 (3.6%)		
**Lifetime psychotic symptoms** Missing: *n =* 4	** *No* **		195 (35.0%)	143 (32.4%)	52 (45.2%)	6.638	**0.012**
	** *Yes* **		362 (65.0%)	299 (67.6%)	63 (54.8%)		
**Comorbid personality disorders** Missing: *n =* 126	** *None* **		379 (87.1%)	318 (90.7%)	61 (72.6%)	31.576	**<0.001**
	** *Borderline* **		27 (6.2%)	14 (4.0%)	13 (15.5%)		
	** *Narcissistic* **		4 (0.9%)	4 (1.1%)	0 (0.0%)		
	** *Histrionic* **		7 (1.6%)	4 (1.1%)	3 (3.6%)		
	** *Obsessive-Compulsive* **		4 (0.9%)	4 (1.1%)	0 (0.0%)		
	** *Schizotypal* **		2 (0.5%)	1 (0.3%)	1 (1.2%)		
	** *Paranoid* **		1 (0.2%)	1 (0.3%)	0 (0.0%)		
	** *Dependent* **		2 (0.5%)	1 (0.3%)	1 (1.2%)		
	** *Not specified* **		9 (2.1%)	4 (1.1%)	5 (5.9%)		
**Lifetime medical comorbidity** Missing: *n =* 130	** *No* **		251 (58.2%)	209 (59.2%)	42 (53.8%)	0.755	0.447
	** *Yes* **		180 (41.8%)	144 (40.8%)	36 (46.2%)		
**Psychiatric comorbidity** Missing: *n =* 376 48	** *None* **		108 (58.4%)	85 (62.0%)	23 (47.9%)	12.376	**0.014**
	** *Eating Disorders* **		10 (5.4%)	3 (2.2%)	7 (14.6%)		
	** *Generalized Anxiety Disorder* **		55 (29.7%)	40 (29.2%)	15 (31.2%)		
	** *Obsessive-Compulsive Disorder* **		5 (2.7%)	3 (2.2%)	2 (4.2%)		
	** *Others* **		7 (3.8%)	6 (4.4%)	1 (2.1%)		
**Medical Multiple comorbidities** Missing: *n =* 138	** *No* **		304 (71.9%)	258 (74.6%)	46 (59.7%)	6.848	**0.011**
	** *Yes* **		119 (28.1%)	88 (25.4%)	31 (40.3%)		

Legend: Standard deviations for quantitative variables and percentages for qualitative variables are reported in brackets. In bold are statistically significant *p* resulting from χ^2^ or unpaired Student’s *t*-tests. BD: Bipolar Disorder LSD: Lysergic acid diethylamide MDMA: 3,4-methylenedioxymethamphetamine (ecstasy).

**Table 2 diagnostics-12-02215-t002:** Recent (last year) or current clinical variables of the total sample and of the two groups obtained according to the presence of lifetime suicide attempts.

Variables			Total Sample N = 561	Absence of Lifetime Suicide Attempts N = 443 (79.0%)	Presence of Lifetime Suicide Attempts N = 118 (21.0%)	χ^2^/t	*p*
**Age at hospitalization**			47.34 (±14.77)	46.95 (±14.89)	48.79 (±14.28)	1.228	0.221
**Smoking Status** Missing: *n =* 190		**Non-smoker**	177 (47.7%)	143 (48.5%)	34 (44.7%)	0.338	0.607
		**Smoker**	194 (52.3%)	152 (51.5%)	42 (55.3%)		
**Duration of hospitalization (days)** Missing: *n =* 41			12.73 (±8.30)	12.58 (±7.76)	13.28 (±10.02)	0.686	0.494
**Number of cigarettes/day** Missing: *n =* 237			9.35 (±12.07)	9.30 (±12.20)	9.51 (±11.65)	0.133	0.894
**Number of major mood episodes in the last year** Missing: *n =* 128			1.49 (±1.02)	1.44 (±1.02)	1.71 (±1.04)	2.182	**0.030**
**Number of manic episodes in the last year** Missing: *n* = 129			0.98 (±0.67)	0.98 (±0.68)	0.99 (±0.64)	0.110	0.913
**Number of depressive episodes in the last year** Missing: *n =* 130			0.33 (±0.66)	0.27 (±0.61)	0.56 (±0.76)	3.322	**0.001**
**Number of hypomanic episodes in the last year** Missing: *n =* 128			0.21 (±0.57)	0.22 (±0.60)	0.21 (±0.49)	0.111	0.912
**Number of major mood episodes triggered by substance abuse in the last year** Missing: *n =* 175			0.13 (±0.46)	0.12 (±0.46)	0.18 (±0.45)	1.131	0.260
**Type of current episode**	** *Manic* **		440 (78.4%)	368 (83.1%)	72 (61.0%)	26.788	**<0.001**
	** *Major Depressive* **		121 (21.6%)	75 (16.9%)	46 (39.0%)		
**Presence of mixed features** Missing: *n =* 4	** *No* **		378 (67.9%)	308 (70.0%)	70 (59.8%)	4.384	**0.045**
	** *Yes* **		179 (32.1%)	132 (30.0%)	47 (40.2%)		
**Administration of poly-therapy during the last mood episode** Missing: *n =* 158	** *No* **		230 (57.1%)	184 (59.0%)	46 (50.5%)	2.707	0.208
	** *Yes* **		171 (42.9%)	128 (41.0%)	45 (49.5%)		
**Type of last mood episode** Missing: *n =* 196	** *First episode* **		3 (0.8%)	3 (1.1%)	0 (0.0%)	10.398	**0.018**
	** *Major Depressive* **		153 (41.9%)	127 (46.3%)	26 (28.6%)		
	** *Manic* **		151 (41.4%)	104 (38.0%)	47 (51.6%)		
	** *Hypomanic* **		58 (15.9%)	40 (14.6%)	18 (19.8%)		
**Comorbidity with hypothyroidism** Missing: *n =* 63	** *No* **		428 (85.9%)	333 (85.6%)	95 (87.2%)	0.170	0.757
	** *Yes* **		70 (14.1%)	56 (14.4%)	14 (12.8%)		
**Comorbidity with diabetes** Missing: *n =* 64	** *No* **		450 (90.5%)	352 (90.7%)	98 (89.9%)	0.066	0.853
	** *Yes* **		47 (9.5%)	36 (9.3%)	11 (10.1%)		
**Comorbidity with hyper-cholesterolemia** Missing: *n =* 127	** *No* **		328 (75.6%)	271 (77.0%)	57 (69.5%)	2.014	0.198
	** *Yes* **		106 (24.4%)	81 (23.0%)	25 (30.5%)		
**Comorbidity with obesity** Missing: *n =* 152	** *No* **		378 (92.4%)	310 (93.6%)	68 (87.2%)	3.779	0.059
	** *Yes* **		31 (7.6%)	21 (6.4%)	10 (12.8%)		
**Achievement of treatment response in the current episode** Missing: *n =* 40	** *No* **		47 (9.0%)	40 (9.8%)	7 (6.2%)	1.335	0.272
	** *Yes* **		474 (91.0%)	369 (90.2%)	105 (93.8%)		
**Achievement of treatment remission in the current episode** Missing: *n =* 40	** *No* **		175 (33.6%)	137 (33.5%)	38 (33.9%)	0.007	1.000
	** *Yes* **		346 (66.4%)	272 (66.5%)	74 (66.1%)		
**Current treatment with statins** Missing: *n =* 111	** *No* **		427 (94.9%)	341 (94.5%)	86 (96.6%)	0.693	0.447
	** *Yes* **		23 (5.1%)	20 (5.5%)	3 (3.4%)		
**Current treatment with levothyroxine** Missing: *n =* 186	** *No* **		343 (91.5%)	288 (91.7%)	55 (90.2%)	0.158	0.802
	** *Yes* **		32 (8.5%)	26 (8.3%)	6 (9.8%)		
**GAF scores** Missing: *n =* 176			56.96 (±14.32)	58.39 (±13.51)	52.84 (±15.82)	3.118	**0.002**
**YMRS scores** Missing: *n =* 52			20.45 (±10.56)	21.52 (±9.93)	16.60 (±11.82)	4.005	**<0.001**
**HAM-D scores** Missing: *n =* 395			14.56 (±6.61)	14.72 (±6.92)	14.28 (±6.06)	0.420	0.675
**MADRS scores** Missing: *n =* 440			22.53 (±8.65)	22.59 (±9.02)	22.42 (±8.09)	0.107	0.915
**BPRS scores** Missing: *n =* 31			40.94 (±9.19)	41.12 (±9.31)	40.27 (±8.72)	0.904	0.367
**HAM-A scores** Missing: *n =* 458			8.95 (±4.35)	9.54 (±4.28)	8.03 (±4.37)	1.729	0.088

Legend: Standard deviations for quantitative variables and percentages for qualitative variables are reported in brackets. In bold are statistically significant *p* resulting from χ^2^ or unpaired Student’s *t*-tests. BPRS: Brief Psychiatric Rating Scale, GAF: Global Assessment of Functioning, HAM-A: Hamilton Anxiety Rating Scale, HAM-D: Hamilton Depression Rating Scale, MADRS: Montgomery–Asberg Depression Rating Scale, YMRS: Young Mania Rating Scale.

**Table 3 diagnostics-12-02215-t003:** Biochemical parameters of the total sample and of the two groups obtained according to the presence of lifetime suicide attempts.

Variables	Total Sample N = 561	Absence of Lifetime Suicide Attempts N = 443 (79.0%)	Presence of Lifetime Suicide Attempts N = 118 (21.0%)	t	*p*
**Number of white blood cells (10^9^/L)** Missing: *n =* 168	7.61 (±2.68)	7.54 (±2.63)	7.85 (±2.85)	0.940	0.349
**Number of red blood cells (10^12^/L)** Missing: *n =* 158	4.55 (±0.55)	4.57 (±0.55)	4.47 (±0.55)	1.551	0.123
**Hemoglobin (g/dL)** Missing: *n =* 157	13.49 (±1.66)	13.56 (±1.64)	13.24 (±1.71)	1.613	0.109
**Mean corpuscular volume (MCV) (fL)** Missing: *n =* 202	86.47 (±9.04)	86.91 (±6.46)	85.06 (±14.48)	1.142	0.256
**Number of platelets (10^9^/L)** Missing: *n =* 206	246.17 (±67.62)	247.29 (±68.44)	242.61 (±65.21)	0.570	0.570
**Mean platelet volume (fL)** Missing: *n =* 207	11.02 (±4.49)	11.04 (±5.11)	10.96 (±1.04)	0.231	0.818
**Number of neutrophils (10^9^/L)** Missing: *n =* 189	4.05 (±2.50)	4.11 (±2.44)	3.85 (±2.71)	0.778	0.438
**Number of lymphocytes (10^9^/L)** Missing: *n =* 188	2.17 (±0.74)	2.14 (±0.75)	2.28 (±0.73)	1.528	0.129
**Neutrophil-to-lymphocyte ratio** Missing: *n =* 189	2.15 (±1.79)	2.19 (±1.81)	1.99 (±1.72)	0.938	0.350
**Glycaemia (mg/dL)** Missing: *n* = 205	93.76 (±28.62)	94.70 (±29.98)	90.56 (±23.29)	1.360	0.176
**Blood urea (mg/dL)** Missing: *n* = 278	29.82 (±16.89)	29.19 (±12.69)	32.90 (±29.95)	0.842	0.404
**Creatinine (mg/dL)** Missing: *n* = 168	0.85 (±0.37)	0.84 (±0.25)	0.91 (±0.63)	1.088	0.279
**Uric acid (mg/dL)** Missing: *n* = 226	5.19 (±1.91)	5.22 (±1.93)	5.08 (±1.88)	0.536	0.593
**Aspartate aminotransferase (U/L)** Missing: *n* = 334	26.67 (±43.21)	27.20 (±47.03)	24.10 (±14.49)	0.747	0.456
**Alanine aminotransferase (U/L)** Missing: *n* = 259	23.48 (±16.88)	23.78 (±17.56)	22.08 (±13.24)	0.799	0.426
**Gamma-glutamyltransferase (U/L)** Missing: *n* = 271	23.09 (±28.05)	22.65 (±28.99)	25.27 (±22.83)	0.691	0.491
**Total bilirubin (mg/dL)** Missing: *n* = 186	0.53 (±0.35)	0.56 (±0.37)	0.42 (±0.22)	4.181	**<0.001**
**Total plasmatic proteins (g/dL)** Missing: *n* = 239	6.58 (±0.56)	6.58 (±0.58)	6.58 (±0.48)	0.015	0.988
**Albumin (g/dL)** Missing: *n* = 229	4.25 (±0.43)	4.26 (±0.44)	4.21 (±0.41)	0.874	0.384
**Lactate dehydrogenase (U/L)** Missing: *n =* 315	208.65 (±95.68)	210.28 (±100.03)	200.71 (±71.34)	0.734	0.465
**Creatine phosphokinase (U/L)** Missing: *n =* 225	214.85 (±334.02)	223.84 (±327.59)	188.09 (±354.89)	0.790	0.431
**Pseudocholinesterase (PCHE) (U/L)** Missing: *n =* 327	7281.97 (±1727.82)	7301.90 (±1616.44)	7188.12 (±2199.32)	0.314	0.755
**Total cholesterol (mg/dL)** Missing: *n =* 203	174.62 (±41.81)	172.21 (±40.03)	182.84 (±46.74)	2.021	**0.044**
**Serum iron (mcg/dL)** Missing: *n =* 351	83.22 (±39.21)	84.79 (±39.50)	75.61 (±37.38)	1.328	0.190
**Thyroid-stimulating hormone (mcU/mL)** Missing: *n =* 241	2.18 (±2.38)	2.23 (±2.61)	2.03 (±1.42)	0.847	0.398
**C-reactive protein (mg/dL)** Missing: *n =* 426	1.19 (±2.78)	0.98 (±1.79)	2.09 (±5.16)	1.081	0.289

Legend: Standard deviations are reported in brackets. In bold are statistically significant *p* resulting from unpaired Student’s *t*-tests.

**Table 4 diagnostics-12-02215-t004:** Summary of the statistics for the best-fit multivariable logistic regression model applied.

Variables	B	S.E.	Wald	*p*	OR	CI
**Total Bilirubin**	−0.420	0.927	0.205	0.651	0.657	0.107–4.042
**Total cholesterol**	0.003	0.006	0.227	0.634	1.003	0.991–1.015
**Presence of lifetime poly-substance-use disorder**	−1.504	0.508	8.785	**0.003**	1.868	1.089–3.204
**Presence of lifetime psychotic symptoms**	0.670	0.424	2.499	0.114	1.955°	0.851–4.487
**Duration of untreated illness**	0.072	0.042	2.938	0.087	1.074	0.990–1.166
**Presence of current obesity**	0.300	0.939	0.102	0.750	1.349	0.214–8.504
**Type of last mood episode**	NA	NA	1.861	0.394	NA	NA

In this analysis, the dependent variable was the presence of lifetime suicide attempts. B = regression coefficient; NA = not applicable; S.E. = standard error of B; Wald = Wald statistics; OR = odds ratio; CI = confidence intervals. B exponential is reported. In bold are statistically significant *p* (≤0.05).

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
