# Peer review of "Which Clinical and Biochemical Parameters Are Associated with Lifetime Suicide Attempts in Bipolar Disorder?"

_diagnostics, 2022, doi:10.3390/diagnostics12092215_

Round 1

Reviewer 1 Report (Previous Reviewer 2)

Hi Dears

Thanks for this research and paper.

Please explain more a bout it's methodology and exclude/include criteria.

How was diagnosis confirmed(Questionnaire or semi-structure interview) ?

Author Response

Thanks for your request of clarifications.

The diagnosis was made by an expert senior psychiatrist of the clinic staff through a clinical interview according to DSM criteria. Previous research by our group showed a high rate of agreement between clinical diagnosis and the use of diagnostic tools for the diagnosis of mood disorders (doi: 10.1111/bdi.12635). We reported these consideration in the method section.

Regarding exclusion/inclusion criteria, we have already reported the requested information in the text, and we added further information about the limits and biases of a retrospective design.

Reviewer 2 Report (Previous Reviewer 4)

No further comments

Author Response

Thank you.

This manuscript is a resubmission of an earlier submission. The following is a list of the peer review reports and author responses from that submission.

Round 1

Reviewer 1 Report

The manuscript is a retrospective investigation of a series of factors and their relation to lifetime suicide attempts in inpatients. 

1. It is unclear whether biochemical parameters were obtained as universal screening in all patients or some could be available because of clinical findings. That could skew those parameters in the patient sample. 

  2. Are rating scale data obtained routinely in all admissions or were they estimated for the study?

3. Please refrain from using "borderline statistical significance" throughout the manuscript.

4. Missingness is massive for much of the data. The authors should provide reasons for different forms of missing data and perhaps not using everything in the final model. 

5. Multivariate models could be better reported, with more statistical information (OR, CIs etc). It is difficult in the present form to understand contributions of each variable.

5. The use of lifetime attempts instead of recent attempts has been pointed out as a significant limitation since changes could be better identified proximally to the event. Have the authors analyzed data on recent attempts?

Reviewer 2 Report

Hi Dears

1-How was Bipoiar diagnosis?(Questionnaire or Expert Psychiatrist /or both of them).

Psychiatrist's diagnosis alone is not enough.

2- In what phase of the disorders (Mania/Depressin/Mixed ) were the Patients.

3- How Attempt to Suicide is assessed?(Questionnaire or Expert Psychiatrist /or both of them).

4-Why Substanceuse was not  one of excluded criteria?

With best regards

Reviewer 3 Report

The authors aimed to identify the biological and clinical features that are associated with lifetime suicide attempts among persons with bipolar disorder. The topic of this study is relevant. As the authors mentioned, this information is crucial in developing appropriate treatment and strategies for suicide prevention. However, some methodological issues mitigated the potential impact of the present study, as detailed below.

The missing data in this study is the main concern. The authors reported that the total sample of identified patients was 561, however, the majority of the variables consisted of a large number of missing data, for example, approximately half of the patient data were missing (376 in psychiatric comorbidity). This issue may render the validity of the main result.

Besides, the rationale for the selection of the biological variables in this study raises another problem of sweeping concept. What is the rationale for the authors to choose those selected biological variables? The hypothesis should come first before the analysis. Please explain the rationale behind the study analysis.

Overall, this study is of importance, however, the methodology and missing data have limited the generalization and validity of the results.

Reviewer 4 Report

This is a very nice study regarding the clinical and biochemical characterizations of bipolar disorder in relation to its suicidality trends. The data are rich and have meaningful prospects both clinically and scientifically, therefore the paper should be published. However, before that the authors might notice the followings:

1 The Abstract is not well written. The subsection “Methods” should be a little bit detailed, and the subsection “Conclusions” should be more like “conclusion” rather than the repetition of results. The abbreviation “DUI” might be defined when it first appears.

2 The Intro part should be enlarged dearly, telling the reader enough rationales of the study, the clear hypotheses, and the purpose of the study. More background can be added in the Intro.

3 The subtypes of BD, for instance, BD I and BD II, should be highlighted throughout the text. In Methods, the authors might isolate the BD I and BD II, and in Results, the data contrasting the two subtypes of BD can be highlighted. The related clinical and biochemical aspects might be more meaningful when they are BD I/II driven rather than they are depression episode driven, since in that way, the current study can offer evidence of the subtype discrimination (rather than of the episode discrimination) and be more “diagnostic”.

4 The authors have noticed the comorbidity of personality disorders, which is wonderful. However, the authors might tell the reader what the diagnostic process of comorbidity is. The borderline personality disorder, for example, often presents parasuicidality, which might be mingled with the suicidal attempts of the bipolar disorder.

5 The “Limitation” part is nice, but the “Conclusions” part might be elevated scientifically, by telling the reader what is the scientific and clinical perspective of the current study, to make the article more “diagnostic”.